# Inheritance of Resistance to Cry1A.105 in *Helicoverpa zea* (Boddie) (Lepidoptera: Noctuidae)

**DOI:** 10.3390/insects13100875

**Published:** 2022-09-27

**Authors:** Wenbo Yu, Graham P. Head, Fangneng Huang

**Affiliations:** 1Department of Entomology, Louisiana State University Agricultural Center, Baton Rouge, LA 70803, USA; 2Bayer Crop Science, Chesterfield, MO 63017, USA

**Keywords:** corn earworm, *Bacillus thuringiensis*, transgenic crop, Cry1A.105, inheritance

## Abstract

**Simple Summary:**

The corn earworm, *Helicoverpa zea* (Boddie), is a target pest of *Bacillus thuringiensis* (Bt) cotton and maize in North America. This pest has evolved resistance to both Bt crops in the United States, which is a significant threat to their sustainable use for *H. zea* control. Cry1A.105 is a Bt toxin commonly used in commercial transgenic maize hybrids to control above-ground caterpillar pests, such as *H. zea*. The aim of this study is to describe the inheritance of Cry1A.105 resistance in *H. zea*. Genetic-cross studies revealed that the resistance to Cry1A.105 is controlled by a single, autosomal, nonrecessive gene. Data generated from the study will be useful in resistance risk assessment and refining Bt resistance mitigation programs for *H. zea* control.

**Abstract:**

Cry1A.105 is a bioengineered *Bacillus thuringiensis* (Bt) insecticidal protein consisting of three domains derived from Cry1Ac, Cry1Ab, and Cry1F. It is one of the two pyramided Bt toxins expressed in the MON 89034 event, a commonly planted Bt maize trait in the Americas. Recent studies have documented that field resistance of the corn earworm, *Helicoverpa zea* (Boddie), to the Cry1A.105 toxin in maize plants has become widespread in the United States. To investigate the inheritance of resistance to Cry1A.105 in *H. zea*, two independent tests, each with various genetic crosses among susceptible and Cry1A.105-resistant populations, were performed. The responses of these susceptible, resistant, F_1_, F_2_, and backcrossed insect populations to Cry1A.105 were assayed using a diet overlay method. The bioassays showed that the resistance to Cry1A.105 in *H. zea* was inherited as a single, autosomal, nonrecessive gene. The nonrecessive nature of the resistance could be an important factor contributing to the widespread resistance of maize hybrids containing Cry1A.105 in the United States. The results indicate that resistance management strategies for Bt crops need to be refined to ensure that they are effective in delaying resistance evolution for nonrecessive resistance (nonhigh dose).

## 1. Introduction

Cry1A.105 is one of the two *Bacillus thuringiensis* (Bt) crystalline (Cry) insecticidal toxins that are produced in transgenic maize containing the MON 89034 event. These pyramided Cry toxin-maize hybrids were first commercially planted in 2010 to control some above-ground lepidopteran pests, such as the corn earworm, *Helicoverpa zea* (Boddie), in the Americas [1]. Studies have shown that MON 89034 maize hybrids were effective against *H. zea* during the initial years of their commercial use [2,3,4,5]. However, recent studies have documented that field populations of *H. zea* in the United States have evolved resistance to various Bt maize events, including MON 89034 [6,7,8,9,10,11,12]. In the southern U.S., *H. zea* is a cross-crop pest that is targeted by both Bt cotton and Bt maize [13,14], and field populations of *H. zea* have also evolved resistance to Bt cotton in this region [15]. The widespread occurrence of Cry toxin resistance represents a significant threat to the sustainability of Bt crops for *H. zea* control.

Knowledge of the inheritance of resistance is foundational in monitoring, risk assessment, and management of Bt resistance [16]. Key aspects of resistance inheritance include the dominance level, the number of alleles and genes involved, and whether resistance is sex-linked (maternal effects) for a given resistant insect population. These characteristics of resistance impact insect resistance management (IRM) programs for Bt crops, such as the adoption of the ‘high dose plus refuge’ strategy [17]. For these reasons, numerous studies have been performed to investigate the inheritance of Bt resistance in many insect pest species targeted by Bt crops. However, due to the well-known difficulty in selection and maintenance of Bt-resistant *H. zea* colonies in the laboratory [18,19], inheritance of resistance to Bt toxins in *H. zea* has been fully studied in only two cases, one with Vip3A resistance and another with Cry2Ab resistance in Texas populations [20,21]. In addition, a recent genome-wide mapping study analyzed the genetic basis of resistance to Cry1Ac in *H. zea* [22].

By using an F_2_ screening method [12], a population of *H. zea* resistant to Cry1A.105 was isolated from insects collected in 2019 in Louisiana. The resistant population has been shown to possess resistance alleles to survive and complete its life cycle (neonate-to-pupa) on single-gene *cry1A.105* maize ears [12]. The availability of the resistant population provided an opportunity to characterize the inheritance of resistance to Cry1A.105 in *H. zea*. In this study, two independent tests, each with various genetic crosses among susceptible and resistant populations, were performed. Responses of the original, crossed, and backcrossed populations to Cry1A.105 were assayed using a diet overlay method. The inheritance of resistance to Cry1A.105 was then analyzed based on these larval responses. Here, we report the results of this first study on the inheritance of Cry1A.105 resistance in *H. zea* and discuss the implications of IRM. 

## 2. Materials and Methods

### 2.1. Sources of H. zea Populations

A single-toxin Cry1A.105-resistant population of *H. zea*, hereafter named Cry1A.105-RR, was established from iso-line families in an F_2_ screen that survived against Cry1A.105 but had no survival against Cry2Ab2 and Vip3Aa20 [12,19]. The Bt-resistant population was able to survive and complete its life cycle on the ears of transgenic plants containing the *cry1A.105* gene [12]. The Cry1A.105-RR and a known Bt-susceptible population (BZ-SS) were used as the original insect sources to examine the inheritance of resistance to Cry1A.105 in *H. zea*. BZ-SS was obtained from Benzon Research Inc. (Carlisle, PA, USA). It has been documented to be susceptible to Cry1Ab, Cry1A.105, Cry2Ab2, and Vip3A, as well as to maize ears expressing one or more of these Bt toxins [8,9,23,24].

### 2.2. Genetic Crosses

In this study, the inheritance of resistance to Cry1A.105 in *H. zea* was assessed in two independent tests (named Test-I and Test-II). Test-I was conducted right after the original Cry1A.105-RR was established and validated as previously described [12]. In the first test, the original Cry1A.105-RR and BZ-SS were reciprocally crossed to generate two F_1_ hybrid populations (Cry1A.105-F_1a_ and Cry1A.105-F_1b_) (Table 1). Cry1A.105-F_1a_ was produced by crossing Cry1A.105-RR_♂_ and BZ-SS_♀_; and Cry1A.105-F_1b_ was generated by crossing Cry1A.105-RR_♀_ and BZ-SS_♂_. The responses of these four insect populations to Cry1A.105 toxins were examined using diet overlay bioassays.

Prior to being used in Test-I, BZ-SS had been maintained and reared on a meridic diet in the laboratory for many generations, while Cry1A.105-RR was recently established from field collections [12]. There was a concern that the genetic background of BZ-SS could be different from that of Cry1A.105-RR. Crosses between insect populations with different genetic backgrounds that are not related to the Bt resistance might cause hybrid vigor that could confound the assessment of Bt resistance [25]. Thus, in Test-II, to ensure a similar genetic background among the insect populations to be evaluated, the original Cry1A.105-RR was crossed and backcrossed with BZ-SS. A population resistant to Cry1A.105 toxin (Cry1A.105-RR’) was reselected on a diet treated with Cry1A.105 toxin at 10 µg/cm^2^ in the F_2_ and F_3_ generations after the backcrosses, respectively. In each resistance reselection, 5–8 neonates were infested on the diet surface of each cell of the 128-cell international trays (C-D International, Pitman, NJ, USA) for 7 days with a total of 4 trays (or ~2500 to 4000 neonates). Approximately 300 healthy larvae, usually 3rd instars, were then selected from each selection. The methods used in the resistance selection were based on a balance of several factors, including the occurrence characteristics of early larval instars of the insect on maize ears, larval cannibalistic behaviors (especially in the late instars), and costs of labor and Bt toxins. The backcrossed-and-reselected resistant population, Cry1A.105-RR’, along with BZ-SS, were used as the insect sources for genetic crosses in Test-II (Table 1). Three types of crosses were performed in Test-II, consisting of (1) reciprocal crosses between BZ-SS and Cry1A.105-RR’ to produce two F_1_ heterozygous populations (Cry1A.105-F_1′a_ and Cry1A.105-F_1′b_); (2) backcrosses of Cry1A.105-F_1′a_ to BZ-SS to produce a mixed backcrossed population (Cry1A.105-BC’); and (3) sib-mating within Cry1A.105-F_1′a_ and Cry1A.105-F_1′b_ to produce a mixed F_2_ population (Cry1A.105-F_2′_) (Table 1). In Test-II, responses of the six populations (BZ-SS, Cry1A.105-RR’, Cry1A.105-F_1′a_, Cry1A.105-F_1′b_, Cry1A.105-BC’, and Cry1A.105-F_2′_) to Cry1A.105 toxins were analyzed using diet overlay bioassays. 

In each genetic cross mentioned above, approximately 50–60 males of an insect population were mass-crossed with 50–60 females of another population in a 20-L cage (Torrance, CA, USA), and the cages were placed in a culture room with a 14:10 h (L:D) photoperiod at 26 °C and >70% r.h. for mating and oviposition [12]. Eggs from each population were collected daily and stored in airbags. Neonate larvae hatching from each population were used in diet overlay bioassays as described below.

### 2.3. Diet Overlay Bioassays with Cry1A.105 Toxin

A diet overlay bioassay method [26] was utilized to measure the responses of each *H. zea* population to the Cry1A.105 toxin. The Cry1A.105 toxin and a related buffer solution used in bioassays were obtained from Bayer Crop Science (St. Louis, MO, USA). Detailed descriptions of the buffer and the methods for measuring Bt toxin molecular weight, toxin concentration, and purity are presented in reference [27]. Each bioassay for BZ-SS consisted of a series of concentrations of Cry1A.105 toxin (0.001, 0.00316, 0.01, 0.0316, 0.1, 0.316, 1, 3.16, and 10 µg/cm^2^), while an additional concentration of 31.6 µg/cm^2^ was added in each bioassay for all other insect populations. The Bt toxin solutions were prepared in distilled water containing 0.1% Triton X-100 to obtain a uniform spread over the diet surface. In the bioassay, approximately 0.8 mL of a liquid diet (Southland Products, Lake Village, AR, USA) was loaded into each cell of the 128-cell trays using syringes, and then, 50 μL of Cry1A.105 solution was applied to the diet surface in each cell. In addition, each bioassay included a buffer-treated negative control and a 0.1% Triton-treated blank control. After the Bt solution on the diet surface dried, one newly hatched larva (<24 h) of a given insect population was released into each cell. In each bioassay, there were four replications with 16–32 larvae in each replicate. Bioassay trays were arranged in growth chambers at 26 °C, ~50% r.h., and a photoperiod of 8:16 h (D:L). The number of dead larvae and larvae that were ≤2nd instars were recorded 7 days after the neonate release. There were two bioassays for Cry1A.105-RR’, one conducted along with BZ-SS and the two F_1_ hybrid populations and another performed along with Cry1A.105-BC’ and Cry1A.105-F_2′_. For all other insect populations, there was one bioassay for each population. 

### 2.4. Data Analysis

A measurement of ‘practical mortality’ described in reference [28] was used to calculate larval mortality: practical mortality (%) = 100 × (number of dead larvae and number of living larvae that were ≤2nd instars)/total number of larvae assayed. The observed practical mortality for each replication in a bioassay was corrected using the negative control mortality [29]. Probit models [30,31] (SAS PROC PROBIT) were used to calculate the median lethal concentration (LC_50_) values and the corresponding 95% confidence limits (95% CLs) for each bioassay, except for the bioassays with Cry1A.105-RR’. For the bioassays with Cry1A.105-RR’, data collected from the two bioassays were combined, and the combined data were analyzed with the probit model to estimate the LC_50_ value and 95% CI. Based on the concentration-response data from the diet overlay bioassays, four concentrations (1.0, 3.16, 10.0, and 31.6 µg/cm^2^) appeared to be appropriate for discriminating the three insect genotypes (Cry1A.105-RR, -RS, and -SS). Thus, for each of the two tests, a two-way analysis of variance (ANOVA) [30], with insect population and Bt concentration as the two main factors (SAS PROC GLM), was used to analyze the corrected larval mortality data at these four concentrations [31]. Treatment means were separated by LSMEANS tests at the α = 0.05 level. As in the probit analysis, the mortality data for Cry1A.105-RR’ at each Bt concentration were combined across the two bioassays for ANOVA. 

Maternal effects of resistance to Cry1A.105 in *H. zea* were examined by comparing the calculated LC_50_ values and larval mortalities at each discriminating concentration between the two F_1_ populations. Significant differences in LC_50_ values and larval mortalities between the two reciprocal F_1_ populations suggest that resistance was not autosomal. Otherwise, if the LC_50_ values and larval mortalities were similar between the two F_1_ populations, the resistance was considered autosomal, not sex-linked, and without maternal effects. 

Dominance levels of resistance to Cry1A.105 in *H. zea* were estimated in two ways: (1) Stone’s dominance “*D*” value [32] referred to here as a ‘genetical’ dominance level, and (2) (functional) effective dominance ‘D_ML_’ [33]. The Stone’s dominance “*D*” value was estimated as:$$D=\frac{2X_{2}-X_{1}-X_{3}}{X_{1}-X_{3}}$$

Here, X_1_, X_2_, and X_3_ were log (LC_50_) values for resistant homozygotes (Cry1A.105-RR and Cry1A.105-RR’), log (LC_50_) values for heterozygotes (Cry1A.105-F_1_ and Cry1A.105-F_1′_), and log (LC_50_) for susceptible homozygotes (BZ-SS), respectively. The concentrations are log-transformed to ensure that the values are normally distributed. The *D* value usually ranges from −1 to 1: a *D* = 1 indicates that the resistance is dominant; 0 < *D* < 1 indicates that the resistance is incompletely dominant, −1 < *D* < 0 indicates that the resistance is incompletely recessive; *D* = −1 indicates that the resistance is recessive; and *D* = 0 indicates that the resistance is neither dominant nor recessive (semidominant or codominant) [32].

Effective dominance, D_ML_, is the functional dominance level of survival at a given insecticide concentration [33]. In this study, D_ML_ was measured at each of the four selected discriminating concentrations mentioned above using the following formula:$$D_{ML}=\frac{ML_{RS}-ML_{SS}}{ML_{RR}-ML_{SS}}$$

Here, ML_SS_, ML_RR_, and ML_RS_ are practical mortalities of BZ-SS, homozygous resistant populations, and F_1_ heterozygous populations at a discriminating Bt concentration, respectively. The D_ML_ value usually ranges from 0 to 1: D_ML_ = 1 indicates that the resistance is dominant, D_ML_ = 0 indicates that the resistance is recessive, 0 < D_ML_ < 0.5 indicates that the resistance is incompletely recessive, and 0.5 < D_ML_ < 1 indicates that the resistance is incompletely dominant. In estimating D_ML_, because larval mortalities of BZ-SS observed in diet overlay bioassays at 1.0, 3.16, and 10.0 µg/cm^2^ of Cry1A.105 were 100% (see results), the larval mortality of BZ-SS at a Cry1A.105 concentration of 31.6 µg/cm^2^ (which was not included in the bioassays) was also assumed to be 100%. 

Finally, chi-square (χ^2^) tests were performed to determine if observed mortalities in F_2_ and backcross populations at each selected discriminating concentration fit the Mendelian single gene model [34,35]. Expected mortality at a given concentration was calculated based on the following formula: expected mortality = 0.25 RR mortality + 0.5 RS mortality + 0.25 SS mortality for an F_2_ population; or expected mortality = 0.5 RS mortality + 0.5 SS mortality for a backcross population. For the same reason mentioned above, the larval mortality of BZ-SS at the concentration of 31.6 µg/cm^2^ was assumed to be 100% in calculating the expected mortalities of the Cry105-F_2′_ and Cry1A.105-BC’ populations. If the observed data fit the single-gene model, the resistance was considered to be controlled by a single gene (or a few tightly linked genes). Otherwise, if the data did not fit the single-gene model, the resistance was considered to be controlled by more than one gene.

## 3. Results

### 3.1. F_2_ Screen-Isolated Cry1A.105 Resistance Was Autosomal in H. zea

Diet overlay bioassays demonstrated that both Cry1A.105-RR and Cry1A.105-RR’ were highly resistant to Cry1A.105. Probit analysis in Test-I showed that, relative to the known Bt susceptible population (BZ-SS), Cry1A.105-RR had a resistance ratio of 2469-fold (Table 2). In Test-2 with the backcrossed-and-reselected population (Cry1A.105-RR’), the resistant population had a resistance ratio of 740-fold relative to BZ-SS (Table 2). For the two F_1_ heterozygous populations in Test-I, Cry1A.105-F_1a_ had an LC_50_ value of 2.49 µg/cm^2^ with a 95% CI of 1.87–3.35 µg/cm^2^ and Cry1A.105-F_1b_ had an LC_50_ of 1.63 µg/cm^2^ with a 95% CI of 0.88–2.93 µg/cm^2^ (Table 2). The difference in LC_50_ values between the two F_1_ populations was not significant based on their overlapping 95% CLs. In Test-II, the LC_50_ values were 9.29 µg/cm^2^ with a 95% CI of 6.16–16.39 µg/cm^2^ for Cry1A.105-F_1′a_ and 7.04 µg/cm^2^ with a 95% CI of 5.39–9.68 µg/cm^2^ for Cry1A.105-F_1′b_. The difference in the LC_50_ between the two F_1_ populations was also not significant (Table 2). In addition, ANOVA showed that larval mortalities between the two F_1_ populations were similar at each of the four discriminating concentrations and for both Test-I and Test-II (Figure 1 and Appendix A). Thus, both the probit analysis and ANOVA show that the resistance to Cry1A.105 in *H. zea* was autosomal and not associated with sex-linkage or maternal effects.

### 3.2. F_2_ Screen-Isolated Cry1A.105 Resistance Was Nonrecessive in H. zea

The LC_50_ value of the combined F_1_ population in Test-I was 1.73 µg/cm^2^ with a 95% CI of 0.88–2.93. Based on the nonoverlapping 95% CIs, the LC_50_ value of the combined F_1_ population was significantly greater than the LC_50_ of BZ-SS, but it was significantly less than the value of Cry1A.105-RR (Table 2). The estimated Stone’s dominance *D* value of the resistance to Cry1A.105 in Test-I was 0.376, suggesting that the resistance was (genetically) incompletely dominant (Table 3). In Test-II, the LC_50_ value of the combined F_1_ population was 8.04 µg/cm^2^ with a 95% CI of 6.24–10.88, which was significantly greater than the LC_50_ of BZ-SS but was not significantly different from the LC_50_ of Cry1A.105-RR’ based on the overlapping 95% CIs (Table 2). The results indicate that the resistance to Cry1A.105 in *H. zea* in Test-II was closer to dominant than recessive. Additionally, the estimated Stone’s *D* value of the resistance in Test-II was 0.773, again indicating that the resistance was incompletely dominant (Table 3).

For both Test-I and Test-II, the effective dominance levels, D_ML_, of the resistance decreased as the Cry1A.105 concentration increased (Table 3). In Test-I, D_ML_ at 1.0 and 3.16 µg/cm^2^ was 0.557 and 0.507, respectively, indicating semidominant resistance, while at the two higher concentrations, 10.0 and 31.6 µg/cm^2^, D_ML_ decreased to 0.383 and 0.396, respectively, indicating incompletely recessive resistance. In Test-II, D_ML_ at 1.0 and 3.16 µg/cm^2^ was 0.958 and 0.947, respectively, indicating near completely dominant resistance. At 10.0 and 31.6 µg/cm^2^, the dominance levels decreased to 0.850 and 0.678, respectively, indicating incompletely dominant resistance (Table 3).

### 3.3. F_2_ Screen-Isolated Cry1A.105 Resistance Was Monogenic in H. zea

Chi-square tests showed that the observed larval mortality of the Cry1A.105-F_2′_ population fit well (*p* > 0.05) with the expected mortality for the Mendelian monogenic model at three of the four discriminating concentrations (Table 4). In fact, the chi-square test for the only significant difference from the single-gene model, which was observed at 31.6 µm/cm^2^, had a *p*-value of 0.0296, which is close to 0.05. In Test-II, the observed larval mortalities of Cry1A.105-BC’ fit well (*p* > 0.05) with the expected mortalities based on the monogenic model for all four discriminating concentrations (Table 4). Together, our results suggest that Cry1A.105 resistance measured in both Cry1A.105-F_2′_ and Cry1A.105-BC’ was controlled by a single locus. 

## 4. Discussion

Genetic crosses between susceptible and resistant populations of *H. zea* from Louisiana in 2019 indicate that Cry1A.105 resistance was likely inherited as an incompletely dominant, single autosomal gene. Prior to the current study, the inheritance of Cry1A.105 resistance in *H. zea* had not been investigated. However, inheritance of Cry1Ac resistance in the Old-World bollworm, *Helicoverpa armigera*, has been investigated in at least seven studies, including three in China [36,37,38], three in India [39,40,41], and one in Pakistan [42]. *H. armigera* in the Old World is a close relative to *H. zea* in the New World, and it is the primary insect pest of cotton and is targeted by Bt cotton in China, India, and Pakistan. The resistance to Cry1Ac in these *H. armigera* populations was inherited as a single autosomal gene for at least six of the seven studies, but the estimated dominance levels varied greatly, ranging from completely recessive (in one study) to completely dominant. Another study by Akhurst et al. [43] reported that resistance to Cry1Ac in an Australian *H. armigera* population was incompletely recessive. *H. armigera* is also the primary target pest of Bt cotton in Australia. Thus, the inheritance of Cry1A.105 resistance in *H. zea* observed in the current study is similar to the reported resistance to Cry1Ac in *H. armigera*. However, a recent study suggested that a mutation in a quantitative trait locus on chromosome 13 was related to the resistance to Cry1Ac in a field/lab-selected *H. zea* strain, but the resistance was likely controlled by more than one locus [22]. Thus, additional studies are needed to elucidate the detailed genetic basis of resistance to Cry1A.105 in *H. zea*. 

Cry1A.105 is not a natural bacterial toxin; it is a bioengineered toxin consisting of domains I and II from the original Cry1Ab/Cry1Ac, domain III from Cry1F, and the C-terminal domain from Cry1Ac [44]. Binding sites of Cry1A.105 toxins in larval midguts have not been investigated for *H. zea*. Competition binding assays for two other lepidopteran pests, *Ostrinia nubilalis* and *Spodoptera frugiperda*, showed that Cry1A.105 shares binding sites with Cry1Ab, Cry1Ac, and Cry1Fa in those two species [45]. In addition, cross-resistance between Cry1A.105 and other Cry1A toxins (e.g., Cry1Ab and Cry1Ac) has been well-documented in several pest species that are targeted by Bt crops, including *H. zea* [11,19,24]. Thus, it is reasonable to assume that Cry1A.105 may also share binding sites with other Cry1A toxins in *H. zea*, and that the mode of action of Cry1A.105 in *H. zea* may be similar to that observed in *O. nubilalis* and *S. frugiperda*. 

The overall conclusions of the two tests with the original (Test-I) and backcrossed-and-reselected resistant (Test-II) populations were generally consistent. The 3-fold difference in the resistance ratios between Cry1A.105-RR in Test-I and Cry1A.105-RR’ in Test-II was likely due to the higher LC_50_ value for BZ-SS in Test-II versus Test-I. In laboratory bioassays, the observed susceptibility to Bt toxins of insect populations measured at different times often varies considerably. For example, Bilbo et al. [8] reported a 2.6- and 7.7-fold variation in susceptibility of the same BZ-SS strain to Cry2Ab2 and Cry1A.105, respectively, between bioassays conducted in 2017 and 2018. Differences were also observed in our previous bioassays [9,11]. Test-I of the current study was performed approximately 6–7 months before Test-II. Thus, the observed 3-fold difference in the Cry1A.105 susceptibility to BZ-SS between the two tests was not surprising. However, some other differences were also observed between the two tests, especially in the estimated dominance levels. For example, the dominance levels (both Stone’s *D* and D_ML_) estimated in Test-II were consistently greater than the corresponding values in Test-I (Table 4). The reasons for such differences are unknown. In fact, the greater fitness of the F_1_ populations on the Bt diet in Test-II compared to Test-I is contrary to expectations because the genetic background among populations evaluated in Test-II was expected to be more similar than for populations examined in Test-I. In addition, our previous study [19] observed that backcross-and-reselected *H. zea* populations that were resistant to single- or dual-Cry toxins slightly outperformed their corresponding source populations. Additional studies are needed to understand why resistant F_1_ populations generated after backcrossing and reselection improved performance.

As described above, the calculated Stone’s *D* values indicated that the Cry1A.105 resistance in the Louisiana *H. zea* population was incompletely dominant. In addition, the effective dominance levels, D_ML_s, measured at the four selected discriminating concentrations, indicated that the Cry1A.105 resistance was functionally nonrecessive, ranging from incompletely recessive to incompletely dominant. The reported expression levels of Cry1A.105 in MON 89034 maize grains (5.9 µg/g) [1] appeared to be well within the range of the selected discriminating concentrations used in the current bioassays. In addition, Yang et al. [20] reported that the D_ML_ of a Texas Cry2Ab2-resistant population varied from incompletely dominant to incompletely recessive at selected concentrations from 1.0 to 31.6 µg/cm^2^, and the resistance was functionally dominant on Cry1Ab/Cry2Ae cotton leaf tissues. A recent analysis of globally published data showed that all six cases of practical resistance to Bt crops, where dominance levels on Bt plants had been evaluated, were associated with functionally nonrecessive resistance [25]. The functionally nonrecessive nature of the resistance to Cry1A.105 and Cry2Ab2 in *H. zea* observed here and by Yang et al. [20] suggests that Bt maize-producing Cry1A.105 and Cry2Ab2 likely does not produce a ‘high dose’ for either of these toxins. Thus, the nonrecessive nature of the resistance to Cry1A.105 and Cry2Ab2 could be an important factor contributing to the widespread occurrence of resistance in *H. zea* in the southern U.S. These results suggest that effort is needed to establish and implement IRM strategies effective in delaying the evolution of Bt resistance controlled by nonrecessive traits. In addition, data obtained from this study will be useful in refining resistance modeling to mitigate the challenge of Cry toxin resistance in *H. zea*.

## Figures and Tables

**Figure 1 insects-13-00875-f001:**
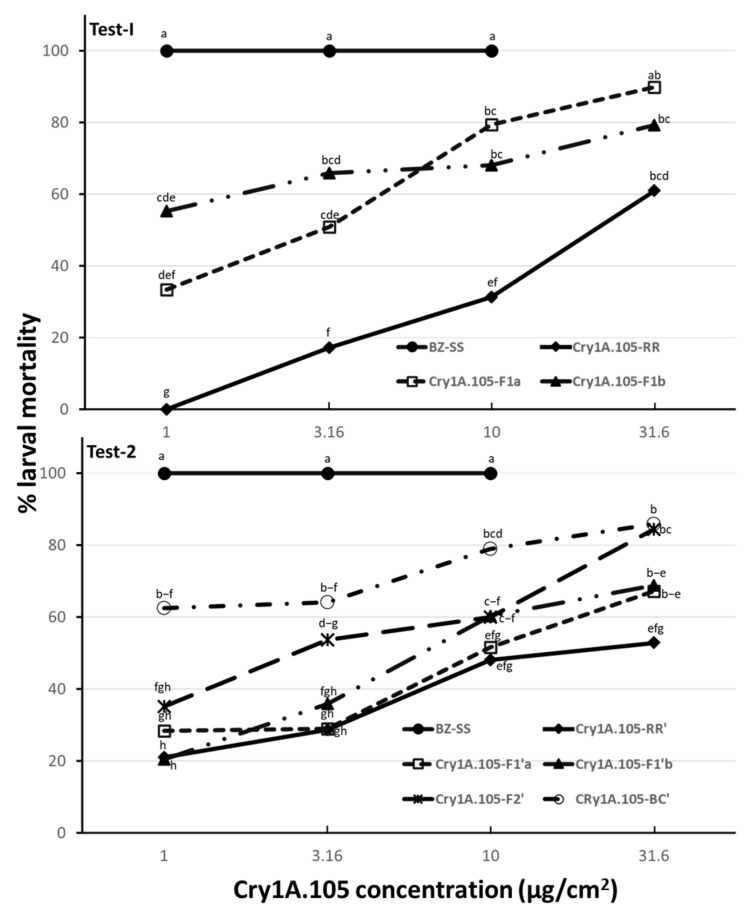
Larval mortalities of different genetic populations of *H. zea* at four Cry1A.105 concentrations in diet overlay bioassays. Means followed by the same letter within each test are not significantly different (LSMEANS tests, α = 0.05). If a mean is followed by four or more letters, an abbreviation with only the first and last letters is presented; for example, ‘b–f’ means ‘bcdef’.

**Table 1 insects-13-00875-t001:** Insect populations used in characterizing the inheritance of Cry1A.105 resistance in *H. zea*.

Insect Population	Source of Insect Population
*Test I with the original Cry1A.105-RR population without backcrosses and reselection*	
BZ-SS	A known Bt susceptible *H. zea* laboratory population provided by Benzon Research Inc (Carlisle, PA, USA).
Cry1A.105-RR	A single-toxin Cry1A.105-resistant *H. zea* population isolated from an F_2_ screen of isoline families collected in a maize field in 2019 in Louisiana.
Cry1A.105-F_1a_	A heterozygous Cry1A.105-resistant population formed by crossing Cry1A.105-RR_♂_ and BZ-SS_♀_.
Cry1A.105-F_1b_	A heterozygous Cry1A.105-resistant population formed by crossing Cry1A.105-RR_♀_ and BZ-SS_♂_.
*Test II with a backcrossed-and-reselected Cry1A.105-resistant strain*	
BZ-SS	A known Bt susceptible laboratory population provided by Benzon Research Inc (Carlisle, PA, USA).
Cry1A.105-RR’	A backcrossed-and-reselected Cry1A.105-resistant *H. zea* population.
Cry1A.105-F_1′a_	A heterozygous Cry1A.105-resistant population formed by crossing Cry1A.105-RR’_♂_ and BZ-SS_♀_.
Cry1A.105-F_1′b_	A heterozygous Cry1A.105-resistant population formed by crossing Cry1A.105-RR’_♀_ and BZ-SS_♂_.
Cry1A.105-F_2′_	A mixed F_2_ population formed by sib-mating within Cry1A.105-F_1′a_ and Cry1A.105-F_1′b_
Cry1A.105-BC’	A mixed population formed from reciprocal back-crosses between (Cry1A.105-F_1′a_ + Cry1A.105-F_1′b_) and BZ-SS.

**Table 2 insects-13-00875-t002:** Concentration-responses of different genetic populations of *H. zea* to Cry1A.105 in diet overlay bioassays.

Insect Population	Number of Larvae Assayed	Slope ± SE	LC_50_(95%CL) (µg/cm^2^)	χ^2^	*p*-Value	Resistance Ratio
*Test-I*						
BZ-SS	620	1.82 ± 0.20	0.008 (0.006, 0.010)	14.75	0.6785	-
Cry1A.105-RR	640	1.50 ± 0.26	19.75 (13.12, 36.97)	22.29	0.0729	2469
Cry1A.105-F_1a_	635	1.20 ± 0.11	2.49 (1.87, 3.35)	26.75	0.2208	311
Cry1A.105-F_1b_	551	0.78 ± 0.12	1.63 (0.88, 2.93)	36.69	0.0118	204
Combined F_1_	1186	0.82 ± 0.10	1.73 (1.10, 2.55)	63.35	0.0061	217
*Test-II*						
BZ-SS	1408	2.75 ± 0.28	0.023 (0.019, 0.028)	28.45	0.0124	-
Cry1A.105-RR’	2285	0.65 ± 0.10	17.01 (10.75, 35.96)	42.99	0.0586	740
Cry1A.105-F_1′a_	1214	0.94 ± 0.12	9.29 (6.16, 16.39)	28.85	0.0503	404
Cry1A.105-F_1′b_	1215	1.04 ± 0.10	7.04 (5.39, 9.68)	24.40	0.1423	306
Combined F_1_	2429	0.99 ± 0.08	8.04 (6.24, 10.88)	54.62	0.0394	350
Cry1A.105-F_2′_	1204	0.56 ± 0.05	2.13 (1.45, 3.32)	29.49	0.2892	93
Cry1A.105-BC’	1088	0.37 ± 0.07	0.10 (0.02, 0.23)	48.68	0.0045	4

**Table 3 insects-13-00875-t003:** Stone’s (*D*) and effective dominance (D_ML_) levels of Cry1A.105 resistance in *H. zea*.

Resistant Population	Calculation Method	Dominance Level	Functional
Cry1A.105-RR	Stone’s *D* value	0.376	Incompletely dominant
	D_ML_ at 1.00 µg/cm^2^	0.557	Codominant
	D_ML_ at 3.16 µg/cm^2^	0.504	Codominant
	D_ML_ at 10.0 µg/cm^2^	0.383	Incompletely recessive
	D_ML_ at 31.6 µg/cm^2^	0.396	Incompletely recessive
Cry1A.105-RR’	Stone’s *D* value	0.773	Incompletely dominant
	D_ML_ at 1.00 µg/cm^2^	0.958	Near completely dominant
	D_ML_ at 3.16 µg/cm^2^	0.947	Near completely dominant
	D_ML_ at 10.0 µg/cm^2^	0.850	Incompletely dominant
	D_ML_ at 31.6 µg/cm^2^	0.678	Incompletely dominant

**Table 4 insects-13-00875-t004:** Testing for monogenic inheritance of Cry1A.105 resistance in Cry1A.105-RR’.

Insect Population	Bt Concentration (µg/cm^2^)	Number of Larvae Assayed	Observed Number of Dead Larvae	Expected Number of Dead Larvae	χ^2^	*p*-Value
Cry1A.105-F_2′_	1.00	128	45	54.3	2.77	0.0961
	3.16	125	67	60.4	1.40	0.2367
	10.0	119	71	77.3	1.47	0.2253
	31.6	64	54	46.2	4.73	0.0296
Cry1A.105-BC’	1.00	128	80	79.6	0.005	0.9436
	3.16	128	82	84.7	0.25	0.6171
	10.0	128	101	99.8	0.07	0.7913
	31.6	64	55	53.8	0.17	0.6801

## Data Availability

The data presented in this study are available on request from the corresponding author.

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
