# Peer review of "Inheritance of Resistance to Cry1A.105 in Helicoverpa zea (Boddie) (Lepidoptera: Noctuidae)"

_insects, 2022, doi:10.3390/insects13100875_

Round 1

Reviewer 1 Report

The study on inheritance of resistance to Bt in insect pest is useful in resistance risk assessment and resistance management. This paper investigated the inheritance of resistance to Cry1A.105 in Helicoverpa zea. The materials and methods, analysis, are explained clearly and at a reasonable level of detail. I have some comments that need help with the article to be clearer. Therefore, I think that the manuscript is worthy of publication.

Line 13:  it should be aim, not arm

Line 61-62, Line 99: the resistance population was established from field-collected individuals in 2019, then the colony was maintained in lab on diet with Cry1A.105 or not?

Line 64 and line 79: the maize ears here only expressing Cry1A.105?

Line 202, here, suggest to add the formular of calculation “the expected dead larvae” in F2 and BC.

Line 221 and 222, change µg/cm2 into µg/cm2

Line 229, Line 247,268, H. zea should be H. zea (italic)

Line 238, LC50 value of the combined F1 population in Test-I was 1.63 µg/cm2 with a 95% CI of 0.88 to 2.93. I think it should be 1.73 (1.10, 2.55). please change it.

Line 259, Table 4, suggest add the” DML “in table of 1.0, 3.16, 10.0, 31.6 (µg/cm2), make it clearer.

Discussion:

1.       The BZ-SS in test I and Test II has so big difference with 2.875 fold, would you like to explain the reasons as possible. If relative to BZ-SS in the test I, the Cry1A.105-RR’ has also 2468.75 fold, nearly equal with the Cry1A.105-RR.

2.       Cry1A.105-RR and Cry1A.105-RR’ colony relative to BZ-SS, Cry1A.105-RR has 2469 fold resistance, and Cry1A.105-RR’ has 740 fold resistance. I wonder is there any relationship between resistance fold and dominance levels. Base on this article, the effective dominance level has lower value with higher resistance fold. From the literature, discuss it more.  

3.       Line 333, I think it should be from incompletely dominant to incompletely recessive at selected concentrations from 1.0 333 to 31.6 µg/cm2

Author Response

Responses to Reviewer #1 comments

 Overall comments: The study on inheritance of resistance to Bt in insect pest is useful in resistance risk assessment and resistance management. This paper investigated the inheritance of resistance to Cry1A.105 in Helicoverpa zea. The materials and methods, analysis, are explained clearly and at a reasonable level of detail. I have some comments that need help with the article to be clearer. Therefore, I think that the manuscript is worthy of publication.

Response: We appreciate the very positive feedback.

Q1. Line 61-62, Line 99: the resistance population was established from field-collected individuals in 2019, then the colony was maintained in lab on diet with Cry1A.105 or not?

R1. Offspring of the F2 families that possessed Cry1A.105 resistance alleles but susceptible to Cry2Ab2 and Vip3A were mixed and reselected on Cry1A.105 diet as described in the reference (Yu et al. 2022 [12]). Test-I was conducted right after the original Cry1A.105-RR was established and validated as described in Yu et al. (2022). To address this question, more detailed information about the time of Test-I and Test-2 has been added in the revised manuscript (Lines 85-87, 313-314).

Q2. Line 64 and line 79: the maize ears here only expressing Cry1A.105?

R2. Yes, the Bt maize is an experimental line, and it expresses only Cry1A.105 protein. It was previously described in the references. No changes are made in the revised manuscript.  

Q3. Line 202, here, suggest to add the formular of calculation “the expected dead larvae” in F2 and BC.

R3. Formula have been added as suggested (Lines 202-205).

Q4. Line 259, Table 4, suggest add the” DML “in table of 1.0, 3.16, 10.0, 31.6 (µg/cm2), make it clearer.

R4. Information was added as suggested (Table 3).

Q5. The BZ-SS in test I and Test II has so big difference with 2.875 fold, would you like to explain the reasons as possible. If relative to BZ-SS in the test I, the Cry1A.105-RR’ has also 2468.75 fold, nearly equal with the Cry1A.105-RR.  

R5. In laboratory bioassays, observed susceptibility to Bt toxins of insect populations measured at different times often varies considerably. For example, Bilbo et al. [8] reported 2.6- and 7.7-fold variation in susceptibility of the same BZ-SS strain to Cry2Ab2 and Cry1A.105, respectively, between bioassays conducted in 2017 and 2018. Similar differences were also observed in our previous bioassays [9,11]. Test-I of the current study was performed approximately 6 to 7 months before Test-II. Thus, the observed 3-fold difference in the Cry1A.105 susceptibility of BZ-SS between the two tests was not surprising. More detailed discussion has been added to address this comment (Lines 306-315).

Q6. Cry1A.105-RR and Cry1A.105-RR’ colony relative to BZ-SS, Cry1A.105-RR has 2469 fold resistance, and Cry1A.105-RR’ has 740 fold resistance. I wonder is there any relationship between resistance fold and dominance levels. Base on this article, the effective dominance level has lower value with higher resistance fold. From the literature, discuss it more.  

R6. I am not aware any relationship between resistance fold and dominance levels has been established. As mentioned in our discussion, in fact, the greater fitness of the F1 populations on Bt diet in Test-II compared to Test-I is contrary to expectations because the genetic background among populations evaluated in Test-II was expected to be more similar than for populations examined in Test-I. We believe additional studies are warranted to reveal the reasons for the better performance of the F1 populations formed from the Bt resistant populations after backcrosses and reselection. The related text was slightly modified to reflect this nature (Lines 364-375).   

Q7. Other minor editorial suggestions

R7. All other editorial changes were made as suggested.

Reviewer 2 Report

Strength of study:

It claims to be the first study on the inheritance of resistance to Cry1A.105 in H. zea and it discusses the implications for Insect Resistance Management (IRM). 

The study is especially crucial as the insect has acquired resistance to insecticides, thus affecting growers of crops on which the insect feeds on  across the world. The manuscript is well written, and the data is well presented. The literature is well cited, the methods used to acquire the data are good. However, minor revisions are needed in the following parts of the manuscript.

Minor revisions:

Line 13: Should it be “aim” instead of “arm”?

Line 99: Should it be “BZ-SS” instead of “SS-BZ”?

Line 192: Should it be “BZ-SS” instead of “SS-BZ”?

Line 240: Should it be “BZ-SS” instead of “SS-BZ”?

Line 245: Should it be “BZ-SS” instead of “SS-BZ”?

Line 227: Italicize “H. zea”.

Line 229: Italicize “H. zea”.

Line 247: Italicize “H. zea”.

Line 268: Italicize “H. zea”.

Table 2: The chi square value “24.4” for Cry1A.105-F1’b should be changed to “24.40” in order to conform to the two decimal places that are running through the entire column.

Author Response

Overall comments: It claims to be the first study on the inheritance of resistance to Cry1A.105 in H. zea and it discusses the implications for Insect Resistance Management (IRM).

The study is especially crucial as the insect has acquired resistance to insecticides, thus affecting growers of crops on which the insect feeds on  across the world. The manuscript is well written, and the data is well presented. The literature is well cited, the methods used to acquire the data are good. However, minor revisions are needed in the following parts of the manuscript..

Response: We appreciate the very positive feedback.

Q1. 11 minor editorial suggestions

R1. All 11 editorial changes were made as suggested

Reviewer 3 Report

I think the manuscript is interesting and that some researchers in the field may be interested to take a look on. I think the most relevant aspect of the article is the conclusion that the resistance to Cry1A.105 in Helicoverpa zea was inherited as a single, autosomal, and non-recessive gene. I think the manuscript may be published in the present form.

Minor comment.

Table 2. H. zea should be written in italics.

Author Response

Overall comments: I think the manuscript is interesting and that some researchers in the field may be interested to take a look on. I think the most relevant aspect of the article is the conclusion that the resistance to Cry1A.105 in Helicoverpa zea was inherited as a single, autosomal, and non-recessive gene. I think the manuscript may be published in the present form...

Response: We appreciate the very positive feedback.

Q1. Table 2. H. zea should be written in italics.

R1. Changed as suggested.